# Detection of Foodborne Viruses in Dates Using ISO 15216 Methodology

**DOI:** 10.3390/v17020174

**Published:** 2025-01-26

**Authors:** Philippe Raymond, Roxanne Blain, Neda Nasheri

**Affiliations:** 1Food Virology National Reference Centre, St. Hyacinthe Laboratory, Canadian Food Inspection Agency (CFIA), 3400 Casavant Boulevard West, St. Hyacinthe, QC J2S 8E3, Canada; 2Food Virology Laboratory, Bureau of Microbial Hazards, Food Directorate, Health Canada, 251 Sir Frederick Banting Driveway, Ottawa, ON K1A 0K9, Canada; neda.nasheri@hc-sc.gc.ca

**Keywords:** human norovirus, hepatitis A virus, murine norovirus, low moisture food, viral extraction

## Abstract

Foodborne viruses such as human norovirus (HuNoV) and hepatitis A virus (HAV) are the major causes of foodborne illnesses worldwide. These viruses have a low infectious dose and are persistent in the environment and food for weeks. Ready-to-eat (RTE) low moisture foods (LMFs) undergo minimal pathogen reduction processes. In recent years, multiple foodborne HAV outbreaks involving hundreds of individuals were associated with the consumption of dates, indicating that they could be important vehicles for foodborne infection. There is no standard method for the extraction and detection of foodborne viruses from dates, but herein we have compared the efficiency of three different protocols based on the ISO 15216 method in the extraction of murine norovirus (MNV) from whole Medjool dates and successfully employed the best performing method in the extraction of HAV, HuNoV GI, and GII and determined the limit of detection (LOD_95_) of 61, 148, and 184 genomic equivalent (gEq) per 25 g, respectively. Finally, we tested the adopted method on various varieties of dates including pitted ones and reported the detection of HuNoV GI and GII from four naturally contaminated date varieties. This ISO 15216 protocol could be employed for surveillance purposes and outbreak management related to dates.

## 1. Introduction

Foodborne viruses such as human noroviruses (HuNoV) and hepatitis A virus (HAV) are highly infectious, persistent in the environment, and resistant to many pathogen reduction strategies [1,2]. They are major causes of nonbacterial foodborne illnesses worldwide [3,4]. In the United States, HuNoV accounts for the most common cause of confirmed, single-etiology foodborne outbreaks (35%) and illnesses (46%) [5]. Over 14% of all foodborne outbreak-related hospitalizations reported in the EU were caused by HuNoV or HAV [6].

Foods can become contaminated with these viruses either at the source of harvest or during food handling and processing [7]. Foodborne viruses maintain their infectivity for a long period of time in food and on surfaces (reviewed in [8]). They are generally resistant to common food preservation strategies including chilling, reduced water activity, and modified atmosphere packaging [9]. Evidence suggests that foodborne viruses can survive desiccation and dry conditions [2,10]. Our previous work has indicated that the storage of foodborne viruses on low moisture foods (LMFs) for 4 weeks leads to less than 1 log reduction in viral infectivity [11]. Due to low infectious dose, environmental stability, and resistance to conventional antimicrobial treatments of foodborne viruses, LMFs contaminated with foodborne viruses can cause serious implications on public health. Several foodborne virus outbreaks were linked to LMFs including dried seaweed and sun-dried tomatoes [12,13].

In recent years, multiple outbreaks of HAV involving hundreds of individuals were associated with the consumption of imported dates (*Phoenix dactylifera*) from the Middle East and North Africa. For example, in 2018, two separate multi-national outbreaks were reported in Denmark and the UK [14,15], in 2019, an outbreak was reported in Sweden [16], and in 2021, another major HAV outbreak was reported in the UK and Australia [17,18]. Date production and processing can introduce contamination through manual handling or the use of contaminated water [19]. Processing can include steps such as cleaning, grading, low-pressure steam heat treatment, sprinkling water and placing under mats in the sun, drying, fumigation, pitting, pasteurization, coating, and packing [19,20]. However, ready-to-eat dates are typically eaten raw or with minimal steam heat treatment during processing [19]. Consequently, in 2023, Food Standards Australia New Zealand (FSANZ) provided a food risk statement indicating that imported RTE dates present a high risk of HAV infection [19].

Currently, there is no standard method for the extraction of foodborne viruses from LMFs such as dates. The International Organization for Standardization (ISO) ISO 15216-1:2017 provides a reference method for the quantitative extraction and detection of HuNoV and HAV from various soft fruits, leaf, stem, and bulb vegetables based on polyethylene glycol (PEG) precipitation [21]. Since the probability of obtaining false-negative results is higher when extraction methods have low recovery rates and/or high inhibition, the ISO 15216-1:2017 standard requires a minimum extraction efficiency of 1% and a maximum RT-qPCR inhibition rate of 75%. The standard also allows the use of alternative RNA extraction kits based on guanidine thiocyanate lysis and silica adsorption, as well as different RT-qPCR detection methods.

We have previously tested the ISO 15216-1:2017 method for the extraction of HAV, HuNoV, or surrogates such as murine norovirus (MNV) and feline calicivirus (FCV), on various matrices [11,22,23,24]. Unfortunately, matrix-associated inhibitory substances, which interfere with the RT-qPCR reaction, are present in some PEG-based RNA extracts [22,23,25]. ISO 15216-1:2017 was not validated for the detection of the target viruses in date matrices and its performance remains to be evaluated.

Herein, we compared the efficiency of two different protocols within the ISO 15216 method for the extraction of MNV from whole Medjool dates; one protocol that is specified for viral extraction from soft fruits, and the other for viral extraction from vegetables. We also evaluated the efficiency of two different RNA extraction kits. We then selected the best performing method for the extraction of multiple foodborne viruses and successfully applied the method for the extraction and characterization of foodborne viruses from different types and varieties of dates and naturally contaminated products. Until now, no outbreaks associated with those positive date samples were reported to our organizations. This method will allow us to conduct future food surveillance and manage potential food outbreaks associated with dates.

## 2. Materials and Methods

### 2.1. Viruses

HuNoV-positive stool samples, HuNoV GI.5 (CFIA-FVR-022, OL345567.1) and GII.4 (CFIA-FVR-020, MT754281.1), were provided by the British Columbia Center for Disease Control (BCCDC). Clarified 10% stool samples were prepared as described previously [22]. HAV HM175 (ATCC® VR-2093) was propagated in the FRhK-4 cell line (ATCC® CRL-1688™) following the American Type Culture Collection (ATCC) recommendation [26]. Murine norovirus-1 (MNV) was provided by Dr. H. Virgin from Washington University (St. Louis, MO, USA) and propagated in the RAW 264.7 cell line, as described previously [27]. Clarified viral suspensions and cultured viral preparations were diluted in aliquots and kept frozen at −80 °C until use. The genomic equivalent (gEq) levels of the virus aliquots were estimated following its extraction using a QIAcube and the RNeasy mini Kit (Qiagen, Mississauga, ON, Canada), as described previously [22]. RNA were eluted in 50 µL RNase-free water, 1 μL of RNasinTM Plus RNase Inhibitor (Thermo Fisher Scientific, Asheville, NC, USA) was added, and the RNA was stored at −80 °C until the RT-qPCR assays, as described below.

### 2.2. Dates Subsamples

Subsamples were prepared with date samples, whole or pitted, collected in various local stores in Quebec. While the targeted weight per subsample was 25 g, which represented 1 to 5 dates, the actual weight range of 22 to 39 g according to the matrix variety yielded an average sample weight of 26 ± 3 g. For instance, two whole Medjool were tested per subsamples, for an average weight of 34 ± 4 g (*n* = 210). The type (whole = W-; pitted = P-), the date varieties (Uns = unspecified), the country of origin (Unk = unknown), the brand (A to T), the sample lot (1 to 3 when multiple), and a unique subsample number were used for sample identification (ex. P-Sayer/Iran/H1-277).

### 2.3. Artificial Contamination (Inoculation)

Unless otherwise specified, virus preparation aliquots were vortexed 2 s and diluted in PBS (Wisent, St-Bruno, QC, Canada) at the targeted gEq level. The diluted aliquots were spiked on different spots (5 × 20 µL) on the surface of the dates in a Whirl–Pak^®^ filter bag (VWR, Mont-Royal, QC, Canada), then left to air dry 30 min in a biosafety cabinet. Non-spiked dates subsamples were included in each extraction batch as negative controls. The amount of virus inoculated was assessed in parallel using the respective RNA extraction method, eGene-up or RNeasy, being evaluated.

### 2.4. Virus Elution and RNA Extraction

The three different ISO 15216 modified approaches tested in the preliminary assays on whole Medjool are schematized in Figure 1. The viruses were extracted using the ISO 15216-1:2017 standard protocol version for soft fruit (ISO-modA) combined with the eGene-up RNA extraction process (Biomérieux, Montréal, QC, Canada), or the vegetable version combined with either the eGene-up (ISO-modB) or the RNeasy kit (ISO-modC) (Qiagen, Mississauga, ON, Canada) to extract the viral RNA following the manufacturer’s instructions. The elution volumes with the eGene-up was 100 µL while the elution volume using the RNeasy kit was 50 µL. Then, 1 µL of RNasin Plus RNase Inhibitor (40 U/µL) (Promega, Madison, WI, USA) was added to the eluate before storage at −80 °C.

### 2.5. RT-qPCR

MNV, HuNoV GI, and HuNoV GII RT-qPCR were carried out as described before [22]. HAV RT-qPCR was performed as described by Larocque et al. [26]. Briefly, RT-qPCR were carried out in a final volume of 25 µL using the TaqMan™ Fast Virus 1-Step Master Mix (Thermo Fisher Scientific, Asheville, NC, USA), and 5 μL of RNA. Primers, probes, and RNA transcripts used in this study can be found in Appendix A. The surfaces, pipettes, and instruments used by the analysts were routinely tested for possible cross-contaminations. Negative no template controls were run on each RT-qPCR. RT-qPCR results were conclusive only when those controls were negative.

### 2.6. Crystal Digital RT PCR (RT-cdPCR)

Crystal Digital PCR was carried on the Naica® System Workflow (Stilla Technologies, Villejuif, France) as described before [26]. Briefly, RT-qPCR were performed using the Quanta Biosciences qScript XLT®One-Step RT-PCR ToughMix® kit (VWR, Mont-Royal, QC, Canada). The reactions were performed in Sapphire chip loaded into the Naica Geode digital PCR for partitioning into 15,000 to 30,000 droplets. Droplet count and quality control for each chamber were performed by the Crystal Reader software while data analysis was performed using the Crystal Miner software version 2.4.0.3 (Stilla Technologies, Villejuif, France). The limit of blanks (LOB) were assessed as described previously on whole Medjool RNA extracts (n = 5) [26]. Positive droplets detected in replicates were considered negative in the analysis when the number of positive droplets was ≤LOB.

### 2.7. Sequencing

Presumptive positive RT-qPCR amplicons were purified on gels, cloned using a TA cloning kit (Thermo Fisher Scientific, Asheville, NC, USA), and sequenced by Sanger, as described before [28]. In one instance (P-Uns/Algeria/I-370), when no bands were visible on gels used for cloning, the RNA extract was amplified by RT-PCR using the RNA Ultrasens kit (Thermo Fisher Scientific, Asheville, NC, USA) as described in ISO 15216 [21]. A sample was confirmed as positive when one aliquot was sequenced as HuNoV or HAV and the sequence was different from the transcript used for the standard curve and the spiked viruses used in the study. The Basic Local Alignment Search Tool (Blast) on NCBI (https://blast.ncbi.nlm.nih.gov/Blast.cgi accessed on 9 April 2024) was used to identify the most similar reference sequence (MSRS).

### 2.8. Recovery Rate Calculation

The recovery rates associated with the virus elution and concentration steps were estimated using the cycle threshold (Ct). The virus recovery rate = 10^(ΔCt/m)^ × 100% where ΔCt is the Ct value of the extracted viral RNA from the matrix minus the Ct value of the viral RNA extracted from the inoculum, and m is the slope of the virus RNA transcript standard curve [21].

### 2.9. Calculation of RT-qPCR Inhibition

Two approaches to assess RT-qPCR inhibition were used. Unless otherwise indicated, RT-qPCR inhibition was estimated using the ratio of the diluted MNV RNA extracted. RT-qPCR inhibition = (1 − 10^(ΔCtdil/m−1)^) × 100% where ΔCtdil is the Ct value of viral RNA extract from the matrix diluted 1/10 in RNase-free water minus the Ct value of the undiluted RNA extracted, and m is the slope of the virus RNA transcript standard curve. Additional tests were also performed using external amplification control (EAC). EAC RT-qPCR inhibition was calculated using the formula = (10^(ΔCt/m)^ × 100%) where ΔCt is the Ct value of viral RNA extract from the matrix minus the Ct value of the inoculum, and m is the slope of the virus RNA transcript standard curve.

### 2.10. Calculation of Limit of Detection (LOD)

Although there is no LOD guideline in the ISO 15216-1:2017 standard, the proportion of positive detections out of three to five replicates for each concentration was used to assess the probability of detection and calculate the LOD_50_ and LOD_95_ with the PODLOD program (v9) [29]. The limit of quantification (LOQ) was defined as the lowest concentration with a coefficient of variation (CV = 100 × SD/mean) below 35%. It was calculated using the back-calculated concentrations of the transcript replicates [30].

### 2.11. Statistical Analyses

One-way ANOVA followed by a Tukey pairwise comparison was used to compare the extraction methodology recoveries (95% CI).

#### 2.11.1. Competition

The impact of viruses’ competition on the recovery rate and LOD estimates was also evaluated. The efficiency of the ISO-modC method to detect HAV, HuNoV GI, and HuNoV GII was assessed by spiking these three viruses together at various concentrations from 10^1^ to 10^4^ on the pitted dates from the Iran variety (P-Uns/Iran/M). The LOD was also assessed as described above.

#### 2.11.2. Robustness

The impact of date varieties or analysts performing the experiments on the ISO-modC method recovery were evaluated using various pitted dates spiked with MNV at 1 to 2 × 10^3^ gEq. Tukey pairwise comparisons were used to compare the MNV recovery results between analysts (95% IC).

#### 2.11.3. Selectivity

The selectivity of RT-qPCR was evaluated in silico by comparing the HAV, MNV, HuNoV GI, and GII primer sequences to the date’s taxon (42345) sequences using the NCBI Blast tool.

## 3. Results

### 3.1. NoV and HAV Recovery Rates from Whole Medjool

For method comparison, we first inoculated the whole Medjool dates with approximatively 10^3^ gEq MNV per 25 g MNV and examined recovery. The MNV recovery yields using the methods without pectinase and chloroform-butanol extraction (ISO-modB and ISO-C) were 3.75 to 4.4 times higher than the one with (ISO-modA) (*p* = 0.002) (Figure 2). The average recovery yields were 39 ± 11% and 44 ± 4% using the ISO-modB and the ISO-modC, respectively. High recovery yields (38%) were also observed with the ISO-modC at lower inoculum concentrations (Table 1). Limited (<16%) to no inhibition was observed with the different approaches. Since the elution volumes using the RNeasy extraction kits was 50 µL compared to 100 µL of the eGENE-up system, the ISO-modC viral RNA concentration was approximately twice the one of the ISO-modB without a higher extraction efficiency. Thus, the ISO-modC was selected for the remaining experiments.

Next, we examined the recovery rates for HuNoV GI, HuNoV GII, and HAV. Using the ISO-modC protocol, the HuNoV GI, HuNoV GII, and HAV recovery rates from whole Medjool range between 36% and 70% (Table 2). These recovery rates are considerably higher than the minimum requirements according to the ISO 15216 method, which is 1%.

### 3.2. Noroviruses and HAV LOD from Whole Medjool

We then measured the limit of quantification (LOQ) for each virus using the transcript standard curves. The LOQ for MNV, HuNoV GI, and GII, and HAV were 0.2, 2.8, 0.8, and 3.3 gEq per µL, respectively. The LOD of the selected ISO 15216-modC protocol for the MNV, HuNoV GI and GII, and HAV spiked on whole Medjool was assessed using both RT-qPCR and RT-cdPCR (Table 3 and Figure 3). The RT-qPCR LOD_50_ and LOD_95_ for the four viruses were below 42 and 184 gEq per 25g, respectively. HAV RT-qPCR LOD_95_ at 61 gEq per 25g (CI95% 30–124) was significantly lower than HAV RT-cdPCR LOD_95_ at 438 gEq per 25g (CI95% 165–1160) (*p* < 0.05). There was no significant statistical difference between the RT-qPCR and RT-cdPCR LOD_95_ for MNV, HuNoV GI, and HuNoV GII. In addition to the number of sample concentrations and replica tested, the accuracy of the LOD estimates and confidence intervals was impacted mainly by the low spiking level. All the non-spiked whole Medjool dates tested as a negative control (MNV n = 12; HAV n = 9; HuNoV GII n = 9; HuNoV GI n = 7) were also negative by RT-qPCR. The Medjool subsamples used for (LOB) were also negative for the four viruses (n = 5). Since RT-qPCR was more sensitive to HAV and as sensitive as RT-cdPCR for the noroviruses, RT-qPCR was selected for the remaining experiments.

### 3.3. Robustness

The robustness of the extraction method to different varieties, brands, or types of dates (pitted or whole) was evaluated by two approaches.

First, nine varieties of pitted and whole dates were spiked with 3 × 10^4^ gEq MNV and were subjected to viral extraction by two different analysts using the ISO-modC method. The MNV recovery rate was above 1% for 33 of the 35 pitted date subsamples tested (Appendix A). The average recovery rate varied from 2.0% to 34.5% (Figure 4). There was no significant difference in the % recovery rate between the two analysts who spiked and proceeded with the extraction 6 months apart (*p* = 0.068). The MNV level of inhibition ranged from −25% to 56% (n = 1) with an average of 17% ± 23% (n = 10) (Appendix A). All those pitted date subsamples were negative for HAV, but six subsamples (17%) from three different varieties were naturally contaminated with either HuNoV GI or HuNoV GII (Appendix A). One subsample from variety P-Uns/Unk/L was positive for HuNoV GI and two for HuNoV GII with Ct at 39.2, 41.9, and 41.7, respectively. Two subsamples from variety P-Uns/Palestine/P were positive for HuNoV GI with Ct at 40.1 and 42.2. One subsample from variety P-Uns/Iran/M was positive for HuNoV GI with Ct of 40.8. All concentrations were below the LOQ. HuNoV GI results were confirmed by sequencing the RT-qPCR amplicons (Table 4). The amplicons from varieties P-Uns/Unk/L, P-Uns/Palestine/P, P-Uns/Iran/M were closest using BLAST analysis to strains typed as HuNoV GI.8, GI.6, and GI.3, respectively. There was no homology to the reference control virus used in this study, which was GI.5 (CFIA-VR-022). Both analysts reported contaminations with different subsamples of # P-Uns/Unk/L, P-Uns/Palestine/P.

Second, different varieties of dates were spiked with HuNoV GI and GII (Table 5). HuNoV GI recovery ranged from 1.1 to 28%, while the HuNoV GII recovery ranged from 1 to 24%. Two non-spiked matrices, an unspecified variety of pitted dates from USA and Algeria (P-Uns/Algeria/I), and a Sayer variety from Iran (P-Sayer/Iran/H), tested positive by RT-qPCR for HuNoV GII. The RT-qPCR amplicon sequences of sample P-Uns/Algeria/I were identical to the control sequence used in the study, suggesting a potential cross-contamination. On the other hand, the P-Sayer/Iran/H sequences were different from the HuNoV GII.4 control (Table 4). The non-spiked pitted Sayer date subsamples (lot A) tested were positive (3/3) for HuNoV GII at 5 ± 3 gEq per g. To verify the extent of the HuNoV GII natural contamination, additional non-spiked pitted Sayer dates from two other boxes of the same lot and boxes from two other lots (Lot B and C) were tested using the ISO-modC protocol and tested by RT-qPCR. All non-spiked Sayer date subsamples tested from three different lots were positive for HuNoV GII and negative for HuNoV GI (n = 14) (Table 6). The HuNoV GII Ct geometric mean was 35.9 and the estimated contamination level was 5 gEq per g (CI95% 3–6). The Sayer lot results were confirmed by sequencing the RT-qPCR amplicons for two lots (A and C). The amplicons shared homology with the HuNoV GII with 3 bp differences over 32 bp compared to the reference control virus used in this study (CFIA-FVR-020) (Table 4).

The HuNoV primers used in the study were the ones described in ISO 15216 and their specificity were verified previously with different human enteric viruses and bacteria [20]. Although HuNoVs were detected in some non-spiked matrix subsamples, sequencing the RT-qPCR amplicons showed there was no matrix cross reactivity using the RT-qPCR detection system with either the whole or the pitted dates tested during the study. In addition, no homology between the primers and the date taxon (42345) was found following NCBI Primer-Blast analysis.

### 3.4. NoV and HAV Recovery Rates and LOD from Pitted Dates

The performances of the ISO-modC protocol on pitted dates was also assessed by spiking HuNoVs in combination with HAV. The recovery rates were all above 1% when HuNoV GI, HuNoV GII, and HAV were spiked simultaneously with more than 290, 1800, and 164 gEq per 25 g, respectively (Table 7).

For the pitted variety P-Uns/Iran/M, the HuNoV GII LOD_95_ estimates at 2200 gEq per 25 g (CI95% 1141–4239) were higher to the level observed with the whole Medjool. On the other hand, both HuNoV GI and HAV LOD_95_ estimates were similar to the level observed with the whole Medjool at 243 gEq per 25 g (CI95% 99–598) and 158 gEq per 25 g (CI95% 74–337), respectively. While the non-spiked controls from P-Uns/Iran/M used in this assay were negative for HuNoV GI (0/4), another subsample P-Uns/Iran/M2-359 of the same lots used when evaluating the MNV recovery assay was positive for HuNoV GI with a Ct 40.8 (1/3). Since this CT value was similar to the subsamples spiked with HuNoV GI at 89 gEq per 25 g (40 to 40.4), the HuNoV GI LOD from this pitted variety might be underestimated since LOD estimates are based on the positive detection ratio (Appendix A).

## 4. Discussion

In recent years, contaminated dates were implicated in multiple foodborne viral outbreaks, but scarce data are available on the ISO 15216 viral extraction method efficiency for this matrix [14,15,16,17,18]. Since dates have high-carbohydrate content, and noroviruses were shown to bind to a variety of carbohydrate moieties, we expected that their extraction from dates would be challenging [31]. For this reason, we compared the efficiency of three ISO 15216 protocols for the extraction of MNV from artificially inoculated Medjool dates and demonstrated that while leafy green ISO-modB and modC versions have comparable efficiencies, ISO-modC, using Qiagen RNA extraction, provides a higher concentration of the viral RNA and thus was selected for further analysis. Furthermore, compared with the soft fruit protocol, ISO-modB and modC have shorter assay times due to the absence of the pectinase and the chloroform-butanol extraction steps, and thus are more cost-effective. Both ISO-modB and modC used RNA extraction methods based on guanidine thiocyanate lysis and silica adsorption, also known as the Boom methodology, and thus achieving similar recovery yields was not unexpected. It would certainly be possible to elute the RNA in 50 µL instead of 100 µL using the eGENEup system and achieve a higher concentration. However, this remains to be investigated as eGENE-up might require a higher volume to saturate the columns compared to the RNeasy system.

The recovery rates achieved on whole Medjool for MNV using ISO-modC were very high (>36%), allowing for the detection of small levels of viral contamination. The HuNoV GI and GII LOD_50_ using RT-qPCR assays were close or below the median dose to achieve 50% infection in secretor-positive subjects reported by Teunis et al. [32] of 2.9 and 95 for GI and GII, respectively. HAV LOD50 was also within the range of the HAV human infectious dose, which is presumed to be in the range of 10 to 100 virions [33]. The ISO 15216-1:2017 protocols were previously used during two HAV outbreaks associated with dates in the UK and Australia but no details on the assay performances were reported [17,18]. Another group based in Denmark compared a modified direct lysis method (Nuclisens, pectinase & Plant RNA Isolation aid) to respond to an HAV outbreak, using the ISO 15216-1:2017 protocol similar to the ISO-modA approach [14,34]. While HuNoV GII and HAV detection remained negative, they reported a low recovery for MNV virus (0.7 to 2%). We reported a higher MNV recovery rate with Medjool dates (8 to 10%) but we did not employ the ISO-modA protocol to the pitted dates. In this study, the ISO-modC performed better compared to ISO-modA with the whole Medjool MNV recovery of 44%, although the recovery from some pitted dates was close to 2%. Rajiuddin and colleagues also reported a high MNV recovery with their optimized direct lysis method (20% to 60%). For this method, they reported an overall processing time for the viral RNA extraction of less than 4 h compared to 6 h and 7.5 h on our side for ISO-modA and ISO-modC, respectively. The estimation of precise hands-on time comparison to their direct lysis method with our ISO-modA and ISO-modC methods is more complicated since we used the eGene-up and the Qiacube robot. Other differences with this study could be related to the differences in the sensivity of the RT-qPCR kits. We previously reported that the TaqMan™ Fast Virus 1-Step was significantly less sensitive to inhibitors in lettuce and berries than the RNA Ultrasens kit [22,23].

The high recovery from whole Medjool was reflected by the quasi absence of inhibition. Meanwhile, the average level of inhibition of the different pitted date varieties was slightly higher, although it was still below the threshold of 75% for all tested varieties. Accordingly, the MNV average recovery yields were lower with the pitted compared to the whole dates but also greatly varied between different varieties. This could be explained by the broken skin of the pitted dates, which leads to exposure of the high-carbohydrate inner parts of the dates that could complicate viral extraction and detection. Furthermore, the type of date, aging, and drying process could influence the level of potential inhibitors and thus the recovery rates.

It is noteworthy that only two subsamples had a recovery below 1% out of 35 tested while the average recovery rate remained above the 1% threshold for all tested varieties. Furthermore, additives such as sunflower oil were present on the ingredient label of some pitted date varieties, which could negatively impact the recovery rates. Thus, the phenol chloroform extraction step could improve the recovery rates for those pitted dates, but this specific use and its impact was not tested in this study.

We selected RT-qPCR over RT-cdPCR to test the different subsamples in this study based on the absence of improvement in terms of sensitivity even when selecting a LOB of 0. LOB could have an important impact on RT-cdPCR LOD estimates compared to RT-qPCR, which is not subjected to such a limit. With our instrument using the ISO-modA protocol to extract RNA from raspberries, the LOB estimates were 2 positive droplets when testing HuNoV GI and HAV and 0 for HuNoV GII [26]. Although all whole Medjool controls remain negative, the presence of low natural contamination background in the pitted ones could complicate the LOB assessment for these matrices. In addition to the better sensitivity, RT-qPCR was superior based on cost (does not require a chip) and user-friendliness. Recently a digital RT-PCR assay for the detection of HuNoV GI and GII RNA in oyster tissue was compared with the ISO 15216 RT-qPCR assays in an interlaboratory study [35]. These authors reported that digital RT-PCR assay could increase uniformity of quantitative results between laboratories; however, they did not assess the potential difference in sensitivity. Digital RT-PCR was reported to be less sensitive to RT-PCR inhibitors associated with the different food matrices but the advantages are less clear in the absence of major inhibition in terms of sensitivity.

Overall, 37 whole and 9 pitted Medjool non-spiked date subsamples were tested during this study over a year and none were positive. This contrasts with the situation observed with some other pitted varieties (P-Sayer/Iran/H, P-Uns/Unk/L, and P-Uns/Palestine/P) where most if not all subsamples, even from multiple lots, were naturally contaminated with HuNoV GI and GII. The increased handling associated with pitted date processing or processing steps specific to those varieties could explain the higher level of natural contamination. The fact that the natural contamination was still detected by two analysts 6 months apart suggests that the RNA is probably protected by its capsid. Repeated HuNoV detection in multiple subsamples and lots over several months, as well as the long shelf life of dates, suggests that pitted dates present a high risk for foodborne viral infection. Diarrhea symptoms that might have been associated with HuNoV infection could be underreported following date consumption since these are well known to have a laxative effect [36]. FSANZ has already identified a high risk of HAV infection from imported ready-to-eat dates, indicating that the processing of this product could be associated with the risk of foodborne pathogen contamination [19].

Both HuNoV GI and GII natural contaminations were detected and confirmed by sequencing. HuNoV GI detection are frequently associated with foodborne transmission while HuNoV GII are more associated with person to person transmission although they both have a high prevalence in water [37]. Dates can be contaminated by the use of water when rinsing the fruits after harvesting [19]. Date pitting machines require water, and a meta-analysis of HuNoV contamination in water sources reported that the global prevalence of HuNoV GI, GII, and GI and GII in water samples was 16.4%, 20.6%, and 12.8%, respectively [38]. The fact that the contaminations were in different types of dates and dates from different origins, and that different NoV genogroups and genotypes were detected, suggests these contaminations were certainly unrelated.

The risk of contamination could be mitigated by different heat treatments such as the one used to accelerate maturation (60 °C steam treatment and 70 °C dryer), dehydration (65 °C tunnel dryers), hydration (low-pressure steam 60 °C), or for glazing (5 min 130°–140 °C or steam treatment 10 min) [20]. It was also shown that heating at 60 °C could reduce viral infectivity [39].

Unfortunately, the short amplicons produced in this study limited our ability for source-tracking. The reported HuNoV GI.8 sequences were relatively rare on GenBank, which further limits the depth of the phylogenetic analysis. On the other hand, there are more than one thousand reported GII sequences with 100% homology to the one detected in the Sayer dates. And, of course, several are also similar to our reference controls. These observations indicate that sequencing the RT-qPCR amplicons could be helpful in excluding false-positive results, but also that sequencing long fragment sizes would help with source-tracking and outbreak investigations.

## 5. Conclusions

ISO-modC, adapted from the ISO 15216 protocol for leafy greens, is efficient at extracting noroviruses and HAV from different varieties of dates. Moreover, the selected protocol allowed for the detection of HuNoVs from several naturally contaminated samples. These contaminations were confirmed by sequencing and point to a high prevalence of HuNoV RNA contamination in dates. Additional information is required to assess the level of risk.

## Figures and Tables

**Figure 1 viruses-17-00174-f001:**
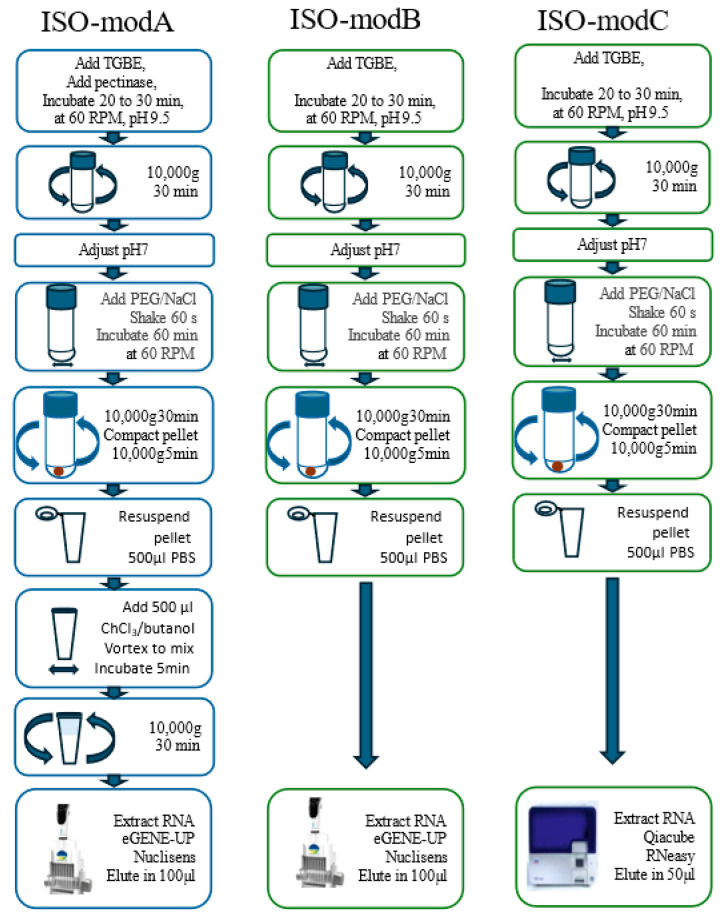
Schematic representation of ISO 15216 protocol modification used in study.

**Figure 2 viruses-17-00174-f002:**
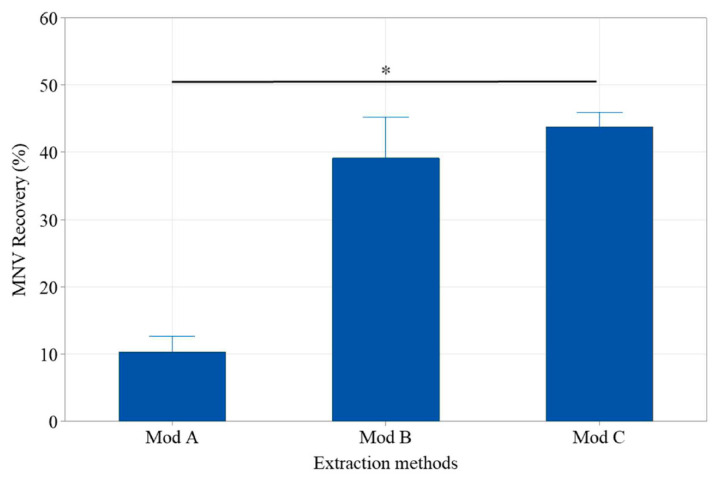
Comparison of MNV recovery efficiency between ISO extraction methods from whole Medjool dates inoculated with 3 log gEq of MNV (n = 3). * The One-Way ANOVA analysis indicates that the recovery rate of the different extraction methods differs significantly (*p* = 0.002).

**Figure 3 viruses-17-00174-f003:**
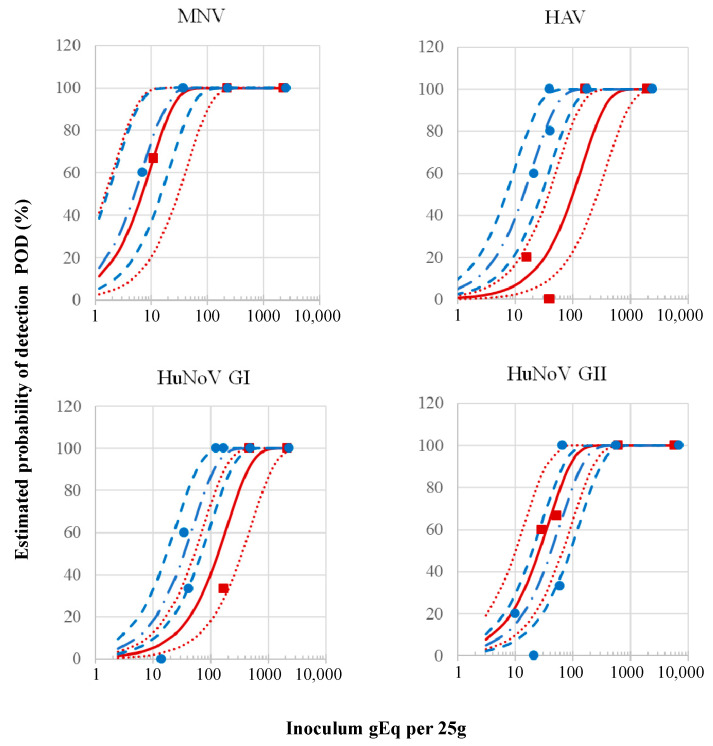
Comparison of estimated probability of detection (POD) curve of MNV, HuNoV. GI, HuNoV GII, and HAV detection methods extracted from whole Medjool using method ISO 15216 modC. Estimated POD curve for viruses spiked dates detected by RT-cdPCR (solid red line) and RT-qPCR (dash–dotted blue line) and 95% confidence band for POD (RTcdPCR dotted lines, RT-qPCR dashed lines). Each observed value represents ratio of positive results from three to five extractions tested using RT-cdPCR (■) and RT-qPCR (●).

**Figure 4 viruses-17-00174-f004:**
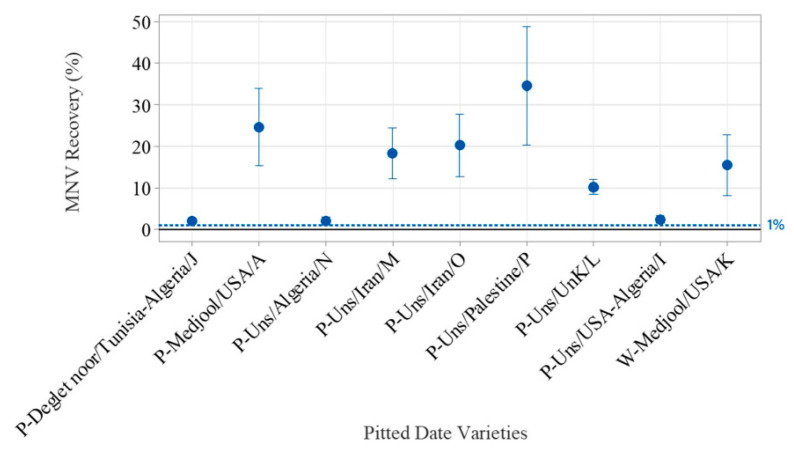
MNV percent recovery yields from various pitted dates using ISO 15216 modC extraction method (average ± sd). Reference blue dotted line = 1% MNV recovery.

**Table 1 viruses-17-00174-t001:** MNV recovery from whole Medjool dates and RT-qPCR inhibition.

Extraction Method	Inoculum Average (gEq per 25 g)	Recovery Average ± sd (%)	Inhibition Average ± sd (%)
ISO15216-modA (soft fruit + eGENE-UP)	2.6 × 10^3^	10 ± 4	nd
	2.3 × 10^4^	10 ± 1	−2 ± 34
	3.3 × 10^5^	8 ± 2	16 ± 11
ISO15216-modB (vegetable + eGENE-UP)	3.9 × 10^3^	39 ± 11	−70 ± 65
	4.0 × 10^4^	39 ± 11	−13 ± 13
	4.4 × 10^5^	27 ± 9	−6 ± 5
ISO15216-modC (vegetable + RNeasy)	2.6 × 10^1^	38 ± 16	nd
	1.6 × 10^2^	38 ± 23	nd
	1.8 × 10^3^	44 ± 4	−7 ± 46

*n* = 3; nd = not detected; inhibition average < 0 = no inhibition.

**Table 2 viruses-17-00174-t002:** HuNoV and HAV RT-qPCR recovery rates from whole Medjool dates using ISO-modC extraction protocol.

Virus	Inoculum Average ± sd (10^3^ gEq per 25 g)	Recovery Average ± sd (%)
HuNoV GI	2.32 ± 0.04	65 ± 13 *^1^
HuNoV GII	7.0 ± 0.3	36 ± 9
HAV	2.4 ± 0.2	70 ± 1

*n* = 3; *^1^ *n* = 5.

**Table 3 viruses-17-00174-t003:** Comparison of limit of detection for MNV, HuNoV GI and GII, and HAV from spiked whole Medjool between RT-qPCR and RT-cdPCR.

Virus	Detection Methodology	LOD_50_ gEq per 25 g (CI95%)	LOD_95_ gEq per 25 g (CI95%)
MNV	RT-qPCR	5 (2–15)	22 (7–66)
	RT-cdPCR	7 (2–31)	30 (7–133)
GI	RT-qPCR	34 (17–68)	148 (75–294)
	RT-cdPCR	134 (52–345)	579 (225–1491)
GII	RT-qPCR	42 (20–91)	184 (86–393)
	RT-cdPCR	26 (10–67)	113 (44–290)
HAV	RT-qPCR	14 (7–29)	61 (30–124)
	RT-cdPCR	101 (38–268)	438 (165–1160)

*n* = 5; CI95% = confidence interval 95%.

**Table 4 viruses-17-00174-t004:** Reference and detected subsample sequences.

Subsample	Varieties	MSRS Genotype	MSRS (% Homology)	5′-3′ Sequences
CFIA-FVR-020	Reference	GII.4	MT754281.1	TCTGAGCACGTGGGAGGGCGATCGCAATCTGGCTCCCAGTTT
CFIA-FVR-022	Reference	GI.5	OL345567	GACCTCGGATTGTGGACAGGAGATCGCAATCTCCTGCCCGAATTC
P-Sayer/Iran/H1-248	Sayer	GII	>1000 sequences	**CT**TGAGCACGTGGGAGGGCGATCGCAATCTGGCTCCCA**A**TTT
P-Sayer/Iran/H3-300	Sayer	GII	>1000 sequences	**CT**TGAGCACGTGGGAGGGCGATCGCAATCTGGCTCCCA**A**TTT
P-Uns/Unk/L-308	Medjool	GI.8	MT372476 (97.8%)	GAC**T**T**A**GG**T**TTGTGGACAGGAGATCGC**G**ATCTC**T**TGCCCGA**T**T**AT**
P-Uns/Palestine/P-358	na	GI.6	MK789655 (97.8%)	GACCT**T**GG**C**TTGTGGACAGGAGATCGCAATCTTCTGCCCGAATTC
P-Uns/Iran/M2-359	na	GI.3	MY492069 (97.8%)	GA**TA**T**GA**G**T**TTGTGGACAGG**G**GA**C**CGC**G**ATCTCCTGCCCGA**T**T**AT**

na = not available. Bold and underlined nucleotides are different from reference genogroup controls.

**Table 5 viruses-17-00174-t005:** HuNoV GI and GII recovery rates with various date varieties and origins.

Type	Varieties	Country of Origin	HuNoV GI		HuNoV GII	
			Inoculum Average gEq per 25 g	Recovery Rate Average ± sd (%)	Inoculum Average gEq per 25 g	Recovery Average ± sd (%)
Whole	Bahri	Jordan	6 × 10^4^	27 ± 2	4 × 10^3^	24 ± 2
Whole	Mazafati	Iran	6 × 10^4^	23 ± 2	4 × 10^3^	16 ± 5
Whole	Deglet Nour	Algeria	3 × 10^4^	16 ± 24	4 × 10^3^	11 ± 15
Whole	Khudari	Saudi Arabia	3 × 10^4^	8 ± 3	4 × 10^3^	6 ± 7
Whole	Zaghoul	Egypt	1 × 10^4^	28 ± 1	1 × 10^4^	23 ± 3
Pitted	Unspecified *^1^	Algeria	1 × 10^4^	1.1 ± 0.2	1 × 10^4^	1.0 ± 1.1
Pitted	Sayer *^2^	Iran	8 × 10^3^	11 ± 3	1 × 10^3^	Inconclusive
Pitted	Unspecified *^2^	USA/Algeria	3 × 10^3^	4 ± 2	2 × 10^3^	Inconclusive

*n* = 3. *^1^ High inhibition in 1 of 3 HuNoV GII subsample tested. *^2^ Control non-spiked dates were negative for HuNoV GI but positive for HuNoV GII. Spiked HuNoV GII recovery results were defined as inconclusive.

**Table 6 viruses-17-00174-t006:** Sayer control subsamples tested using HuNoV GII RT-qPCR.

Subsample	Lot	Target Name	CT	Estimated Target RNA Concentration (gEq per g)
P-Sayer/Iran/H1-248 *^1^	Lot1 box 1	HuNoVGII	34.8	7.3
P-Sayer/Iran/H1-249	Lot1 box 1	HuNoVGII	35.3	5.4
P-Sayer/Iran/H1-250	Lot1 box 1	HuNoVGII	36.6	2.2
P-Sayer/Iran/H1-290	Lot1 box 2	HuNoVGII	38.2	0.8
P-Sayer/Iran/H1-291	Lot1 box 2	HuNoVGII	35.8	3.9
P-Sayer/Iran/H1-292	Lot1 box 2	HuNoVGII	35.8	3.8
P-Sayer/Iran/H1-293	Lot1 box 3	HuNoVGII	37.8	1.0
P-Sayer/Iran/H1-294	Lot1 box 3	HuNoVGII	37.1	1.6
P-Sayer/Iran/H1-295	Lot1 box 3	HuNoVGII	36.6	2.3
P-Sayer/Iran/H2-296	Lot2	HuNoVGII	34.4	9.8
P-Sayer/Iran/H2-297	Lot2	HuNoVGII	34.6	8.7
P-Sayer/Iran/H2-298	Lot2B	HuNoVGII	35.9	3.5
P-Sayer/Iran/H3-299	Lot3	HuNoVGII	35.6	4.4
P-Sayer/Iran/H3-300 *^1^	Lot3	HuNoVGII	34.2	11.0

*^1^ Subsample sequenced.

**Table 7 viruses-17-00174-t007:** HuNoVs and HAV recovery rates from pitted date variety P-Uns/Iran/M when spiked together.

HuNoV GI		HuNoV GII		HAV	
Inoculum Average gEq per 25 g	Recovery Average ± sd (%)	Inoculum Average gEq per 25 g	Recovery Average ± sd (%)	Inoculum Average gEq per 25 g	Recovery Average ± sd (%)
8.9 × 10^1^	15 ± 14 (3/5)	1.0 × 10^3^	10 ± 9	8.8 × 10^1^	23 ± 11 (4/5)
2.9 × 10^2^	10 ± 6	1.1 × 10^3^	2 ± 2 (2/5)	8.3 × 10^0^	nd (0/5)
6.8 × 10^2^	11 ± 11	1.8 × 10^3^	10 ± 10	1.64 × 10^2^	24 ± 19
2.7 × 10^3^	7 ± 3	1.1 × 10^4^	6 ± 2	2.7 × 10^2^	7 ± 1

*n* = 5; nd = not detected. All non-spiked control subsamples were negative to tested viruses (0/4). Parentheses are positive results per total subsample tested; detected sample status is 5/5 unless otherwise specified.

## Data Availability

The datasets generated during and/or analyzed during the current study are available from the corresponding author on reasonable request.

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
