# Peer review of "Detection of Foodborne Viruses in Dates Using ISO 15216 Methodology"

_viruses, 2025, doi:10.3390/v17020174_

Round 1

Reviewer 1 Report

Comments and Suggestions for Authors

The manuscript “Detection of Foodborne Viruses in Dates Using the ISO 15216 2 Methodology” describes an interesting in-house validation work of a method for the detection of HAV, NoVGI and NoVGII in dates. Overall the work and the papers are well conceived. However, some points should be addressed before acceptance for publication.

General comments:
1) The Authors should clarify why their experimental plan combined the two RNA extraction methods (eGENE-up and RNAeasy) only with the leafy green vegetables protocol of ISO (unless tests for the different extractions were run separately and the the significant difference between modA and modB excluded that the use of RNAeasy could provide any significant improvement). Also, it would also be clearer for the reader if you included in Fig. 1 the two PCR (real-time, crystal digital) approaches tested in the study.

2) Methods, Selectivity section (2.11.3) and matrix cross reactivity (lines 330-332): the absence of match between the primers and the matrix itself should not be considered as selectivity, as this should comprise any other sequence that may be present in the sample (e.g. microbiota, nucleic acids from other human sources, etc).  Please revise.

3) The Results section is very difficult to follow.  We suggest to shorten the text, focusing it on the key results (and referring to the tables for details as Ct values for specific samples.  Also, the Authors report the quantitative results for NoV detection I samples as gEq per 50 microL of extracts. This format makes comparison of results to other studies difficult and the Authors should consider reporting values as genome copies or genome equivalents per microL, as per ISO standard.

4) The discussion should address more extensively the comparison of the results with the paper from Rajiuddin et al (2020).  As it is, the Authors only comment on comparison of method modA to the results from the other group, but do not discuss the results of their optimized method (modC) to the lysis method, for example, considering the overall time of the two procedures, their recoveries, etc

Specific comments:

- lines 97-98: this sentence is unclear. Why “notably”?  What is the relevance of this?

- lines 105-106: it is unclear how the virus spotted on dates could dry to air if already inserted in the Whirl-Pak.  Did the Authors take special precautions to ensure complete attachment of the virus to the dates? In our lab experience on dates, full drying of a smaller volume (10 microL) of viral stock spotted on dates required 30 min fully exposed to air flow in the cabinet on Petri dish.  Please clarify.

- line 124: check the format of the reference

- Sequencing section (2.7): please clarify in the text why sequencing of the real-time amplicon was chosen for confirmation and characterization of naturally contaminated samples, in place of the most common sequencing of partial capsid region

- Line 165: please clarify why a different number (3 or 5) of replicate was adopted in the LOD calculation, as this affect the confidence intervals.

- Competition section (2.11.1): this section is a bit unclear. If I understood correctly, with this experiment you are addressing the effect of simultaneous spiking of a sample with more than one virus on the recovery.  If so, please state it at the beginning of the paragraph.

- Results, lines 197-198 and discussion: the Authors should address in the discussion the fact that, in presence of a 2-fold factor in the elution volume, method modC provides a higher recovery but not a 2-fold higher recovery.  Therefore, this RNA extraction procedure provides a more concentrate RNA but not a higher extraction efficiency.

- Tables’ footnotes: please check the character of the tables’ footnotes as it is larger than the main text size and the current formatting of text/tables is a bit confounding

- Line 219: I would suggest to avoid expressing the LOD “collectively” for the four viruses.  If you want to express the highest boundary in the LODs calculated, better to report is as “the highest LOD50 and LOD95 was … for the PCR targeting ….”

- Results of sequencing (Table 5): as the real-time region sequenced is not highly discriminating, we suggest to remove these results from the main text and move them to Supplementary and only report results in the text as putative identifications.

- Supplementary: It is unclear why MNV testing is reported for 1/10 dilution

Author Response

The manuscript “Detection of Foodborne Viruses in Dates Using the ISO 15216 2 Methodology” describes an interesting in-house validation work of a method for the detection of HAV, NoVGI and NoVGII in dates. Overall the work and the papers are well conceived. However, some points should be addressed before acceptance for publication.

General comments:
Comments 1) The Authors should clarify why their experimental plan combined the two RNA extraction methods (eGENE-up and RNAeasy) only with the leafy green vegetables protocol of ISO (unless tests for the different extractions were run separately and the the significant difference between modA and modB excluded that the use of RNAeasy could provide any significant improvement). Also, it would also be clearer for the reader if you included in Fig. 1 the two PCR (real-time, crystal digital) approaches tested in the study.

Response 1)

Thank you for the suggestions. Both RNA extraction methods compared in ISO-modB and modC are based on the Boom methodology as required in ISO 15216. As such we expected little variation in terms of recovery yields between the two methods. The different extraction tests were performed separately. According to the ISO 15216 standard, the threshold to provide acceptable results is only 1%, and that we reached 36%, thus we did not expect a significant improvement with the ISO-modA if the Qiagen kits were used. Nevertheless, as indicated in line 387 there are still items to study regarding ISO-modA recovery rates including from pitted dates for those matrices which have low recovery with the leafy green approach:

Page 14, Line 387: Thus, the phenol chloroform extraction step could improve the recovery rates for those pitted dates, but this specific use and its impact was not tested in this study.

Regarding the Fig1, we think it does reflect the ISO 15216 protocol modifications. The standard requires the use of real-time RT-PCR only, although different real-time RT-PCR kit could be used.     

We added the following details to the text.

Page 2, Line 67. The standard also allows the use of alternative RNA extraction kits based on guanidine thiocyanate lysis and silica adsorption, as well as different RT-qPCR detection methods.

Page 12, Line 344. Both ISO-modB and modC used RNA extraction methods based on guanidine thiocyanate lysis and silica adsorption, also known as the Boom methodology, and thus achieving similar recovery yields was not unexpected.

Comments 2) Methods, Selectivity section (2.11.3) and matrix cross reactivity (lines 330-332): the absence of match between the primers and the matrix itself should not be considered as selectivity, as this should comprise any other sequence that may be present in the sample (e.g. microbiota, nucleic acids from other human sources, etc).  Please revise.

Response 2)

We agree with the reviewer that the absence of match between the primers and the matrix itself is not sufficient to be considered as selectivity. The HuNoV primers used in our study are not new and are described in the ISO 15216 standard. They have been validated and their specificity verified previously. “The specificity of the primers was verified with six different human enteric viruses: poliovirus (serotype 1 vaccine strain); HAV; hepatitis E virus; Aichi virus; astrovirus; and rotavirus. The specificity was also tested on seven bacteria that could be detected in BMS: Escherichia coli, Shewenella putrefaciens, Chromobacterium violaceum, Aeromonas sobria, Vibrio alginolyticus, Vibrio parahaemolyticus and Vibrio cholerae. None of the tested viruses or bacteria gave positive results “. Nevertheless, we wanted to point out that, with the date matrix, the amplification of the virus in non-spiked matrices was not a cross-reactivity result.

In the revised version of the manuscript, we added some information on the primer specificity:

Page 8, Line 255.  3.3. Robustness

Page 11, Line 306. The HuNoV primers used in the study were the ones described in the ISO 15216 and their specificity have been verified previously with different human enteric viruses and bacteria 20. Although HuNoVs were detected in some non-spiked matrix subsamples, sequencing the RT-qPCR amplicons showed there was no matrix cross reactivity using the RT-qPCR detection system with either the whole or the pitted dates tested during the study. In addition, no homology between the primers and the dates taxon (42345) was found following NCBI Primer-Blast analysis.

Comments 3) The Results section is very difficult to follow.  We suggest to shorten the text, focusing it on the key results (and referring to the tables for details as Ct values for specific samples.  Also, the Authors report the quantitative results for NoV detection I samples as gEq per 50 microL of extracts. This format makes comparison of results to other studies difficult and the Authors should consider reporting values as genome copies or genome equivalents per microL, as per ISO standard.

Response 3)

The results section was revised to address some of the reviewer concerns. We estimated that the Ct information in the text from non-spiked sample is important in many instances to reflect the low level of contamination and would not be clear if we used <0.04 gEq per g. To avoid the duplication of information with the text result, we move Table 4 describing the Ct of non-spiked samples to the Supplementary file (S2).

We made some correction to the units.

The format gEq per 50 µl was used to reflect the total number of viruses detected per sample extract. The format gEq per 25 g reflects the total number of virus inoculated per sample. In our view these format better reflects the risk associated with the matrix, for instance when compared to Teunis et al 31. We do not used the gEq per 25 g unit for the none-spiked samples since we do not have each virus recovery rates for each date varieties. We agree with the reviewer’s suggestion to use the ISO format gEq per g for the non-spiked samples. We also corrected the unit of the LOQ to gEq per µl since the LOQ is calculated using the RNA transcript values.

Page 7 Line 230. The LOQ for MNV, HuNoV GI and GII, and HAV were 0.2, 2.8, 0.8 and 3.3 gEq per µl, respectively.  

Page 9, Line 278. Table 4 was moved to the supplementary files and renamed as Supplementary Table S2.

Page 8, Line 260.  Supplementary Table S2.

Page 8, Line 264. Supplementary Table S3

Page 9 Line 290. The non-spiked pitted Sayer date subsamples (lot A) tested were positive (3/3) for HuNoV GII at 5 ± 3 gEq per g.

Page 9 Line 296. The HuNoV GII Ct geometric mean was 35.9 and the estimated contamination level was 5 gEq per g (CI95% 3-6).

Table 6. The values were modified from gEq 50 µl of sample extract to gEq per g

The table 5,6,7,8 numbers were adjusted.

An error was detected between table 7 and the text. The unit of the spiked samples inoculum were corrected from gEq per RT-qPCR to gEq per 25 g page 11 line 322, line 325 and line 329. Also, for precision, the text was modified to clarify that the potential LOD underestimation (not over) is limited to the sample tested.

Page 11, Line 329. Since this CT value was similar to the subsamples spiked with HuNoV GI at 89 gEq per 25g (40 to 40.4), the HuNoV GI LOD from this pitted variety might be underestimated since LOD estimates are based on positive detection ratio.

Comments 4) The discussion should address more extensively the comparison of the results with the paper from Rajiuddin et al (2020).  As it is, the Authors only comment on comparison of method modA to the results from the other group, but do not discuss the results of their optimized method (modC) to the lysis method, for example, considering the overall time of the two procedures, their recoveries, etc

Response 4)

In the revised version, we presented both the ISO-modC and Rajiuddin optimized method recovery results page 12, line 363 and 365. Although we have not tested their lysis method, we added some discussion on overall processing time since they reported the information.

Page 12, Line 365. Rajiuddin and colleagues also reported a high MNV recovery with their optimized direct lysis method (20% to 60%). For this method, they reported an overall processing time for the viral RNA extraction of least than 4 h compared to 6 h and 7.5 h on our side for ISO-modA and ISO-modC, respectively. The estimation of precise hands-on time comparison to their direct lysis method with our ISO-modA and ISO-modC is more complicated since we used the eGene-up and the Qiacube robot.

Specific comments:

Comments - lines 97-98: this sentence is unclear. Why “notably”?  What is the relevance of this?

Response- lines 97-98:

Owing to the nature of the product, the weight of the whole Medjool samples tested were higher than the targeted weight. One could expect the recovery yields to decrease with more matrix per sample. But this interpretation remains speculative, and the sentence remains only informative.

We changed “Notably” for “For instance”:

Page 3, Line 106. For instance, two whole Medjool were tested per subsamples, for an average weight of 34 ± 4 g (n=210).

Comments - lines 105-106: it is unclear how the virus spotted on dates could dry to air if already inserted in the Whirl-Pak.  Did the Authors take special precautions to ensure complete attachment of the virus to the dates? In our lab experience on dates, full drying of a smaller volume (10 microL) of viral stock spotted on dates required 30 min fully exposed to air flow in the cabinet on Petri dish.  Please clarify.

Response - lines 105-106:

Indeed, we took special precautions when spiking food matrices. The 100 µl of viruses were spiked in multiple spots (5 x 20 µl) on the date surfaces in the mesh filter bag which are left open under the biosafety cabinet air flow. 

We added the following details:

Page 3, Lines 113. The diluted aliquots were spiked on different spots (5 x 20 µl) on the surface of the dates in a Whirl–Pak® filter bag (VWR, Mont-Royal, QC, Canada), then left to air dry 30 min in a biosafety cabinet.

Comments - line 124: check the format of the reference

Response- line 124:

We corrected the format of the references:

Page 4, Line 133. The HAV RT-qPCR was performed as described by Larcoque et al. 26

Page 12, Line 351. The HuNoV GI and GII LOD50 using the RT-qPCR assays were close or below the median dose to achieve 50% infection in secretor-positive subjects reported by Teunis et al. 32 of 2.9 and 95 for GI and GII, respectively.

Comments - Sequencing section (2.7): please clarify in the text why sequencing of the real-time amplicon was chosen for confirmation and characterization of naturally contaminated samples, in place of the most common sequencing of partial capsid region.

Response - Sequencing section (2.7):

The sequencing of amplicon is a standard process used by the CFIA diagnostic laboratories to survey food and confirm RT-qPCR results. It allows the exclusion of false-positive results associated to cross-contamination, for instance with the reference transcript with insert used for quantification. While the information on the RT-qPCR method using RNA transcript with insert is described Page 4, Line 132 and in the cited references 22 and 26, we added some information on the RNA transcripts and their sequences.

Page 4, Line 135. Primers, probes and RNA transcripts used in this study can be found in Supplementary Table S1.

Page 15, Line 466. Table S1. List of primers, probes, and RNA transcripts

Supplementary Table S1. List of primers, probes, and RNA transcripts

Comments - Line 165: please clarify why a different number (3 or 5) of replicate was adopted in the LOD calculation, as this affect the confidence intervals.

Response  - Line 165:

There is no minimum or maximum LOD requirement in the ISO 15216. There is no guideline on the number of replicates as well. The confidence interval is impacted mainly by the number of concentrations tested in the fraction section. LOD50 for the four viruses with both RT-qPCR and RT-cdPCR was very low <134 gEq per 25 g. Spiking various concentrations at those concentration, close to 1-2 gEq per µl, is by itself a challenge in terms of accuracy that have a major impact on the confidence interval.

We added the following information:

Page 5, Line 179. Although there is no LOD guideline in the ISO 15216-1:2017 standard, the proportion of positive detections out of three to five replicates for each concentration was used to assess the probability of detection and calculate the LOD50 and LOD95 with the PODLOD program (v9)29.

Page 7, Line 238. In addition to the number of sample concentrations and replicas tested, the accuracy of the LOD estimates and confidence intervals were impacted mainly by the low spiking level.

Comments - Competition section (2.11.1): this section is a bit unclear. If I understood correctly, with this experiment you are addressing the effect of simultaneous spiking of a sample with more than one virus on the recovery.  If so, please state it at the beginning of the paragraph.

Response - Competition section (2.11.1):

The sentence at the beginning of the paragraph was revised as follows:

Page 5, Line 189. The impact of viruses’ competition on the recovery rate and LOD estimates was also evaluated.

Comments - Results, lines 197-198 and discussion: the Authors should address in the discussion the fact that, in presence of a 2-fold factor in the elution volume, method modC provides a higher recovery but not a 2-fold higher recovery.  Therefore, this RNA extraction procedure provides a more concentrate RNA but not a higher extraction efficiency.

Response- Results, lines 197-198 and discussion:

We modified the result description the following.

Page 6, Line 212. Since the elution volumes using the RNeasy extraction kits was 50 µl compared to 100 µl the eGENE-up system, the ISO-modC viral RNA concentration was approximately twice the one of the ISO-modB without a higher extraction efficiency.

Comments - Tables’ footnotes: please check the character of the tables’ footnotes as it is larger than the main text size and the current formatting of text/tables is a bit confounding

Response - Tables’ footnotes:

We presume the font style and size will be selected by the editor. Nevertheless, the font size of the table footnotes of Table 1, 2, 3, 5 and 6 were modified to 10.

Comments - Line 219: I would suggest to avoid expressing the LOD “collectively” for the four viruses.  If you want to express the highest boundary in the LODs calculated, better to report is as “the highest LOD50 and LOD95 was … for the PCR targeting ….”

Response- Line 219:

The collective description describes the similarity between the LOD. The detailed numbers are available for the readers in the table. We prefer to keep the collective description to shorten the text and facilitate the result comprehension.

Comments - Results of sequencing (Table 5): as the real-time region sequenced is not highly discriminating, we suggest to remove these results from the main text and move them to Supplementary and only report results in the text as putative identifications.

Response- Results of sequencing (Table 5):

We respectfully disagree and believe that the sequences identified here could be very useful. The sequence diversity facilitates the comprehension of what we describe as unrelated contaminations.

Comments - Supplementary: It is unclear why MNV testing is reported for 1/10 dilution

Response- Supplementary:

Two approaches to assess PCR inhibition was used with the pitted and whole dates described in Table S3. As described in the Material and method section, since the dates were spiked with MNV, the inhibition could be estimated by testing MNV in the undiluted and 1/10 diluted RNA extract. The inhibition could also be calculated by spiking an external amplification control (EAC) in the undiluted RNA extract. The EAC tested were HuNoV GI, GII and HAV.

We added some information on the EAC and MNV 1/10 in the Materials and Methods and in the Supplementary Table S3.

Page 5, Line 168. Two approaches to assess RT-qPCR inhibition were used. Unless otherwise indicated, the RT-qPCR inhibition was estimated using the ratio of diluted MNV RNA extracted.

Page 5, Line 173. Additional tests were also performed using external amplification control (EAC). The EAC RT-qPCR inhibition was calculated using the formula = (10(ΔCt/m) × 100%) where ΔCt is the Ct value of viral RNA extract from the matrix minus the Ct value of the inoculum, and m is the slope of the virus RNA transcript standard curve.

Page 15, Line 468. Table S3. RT-qPCR Inhibition rate from pitted and whole dates.

Supplementary Table S3. RT-qPCR inhibition rates from pitted and whole dates.

*2 The MNV RT-qPCR inhibition was estimated using the ratio of diluted 1/10 MNV RNA extracted. The RT-qPCR inhibition = (1-10(ΔCtdil/m-1) ) × 100% where ΔCtdil is the Ct value of viral RNA extract from the matrix diluted 1/10 in RNase free water minus the Ct value of the undiluted RNA extracted, and m is the slope of the virus RNA transcript standard curve.

Reviewer 2 Report

Comments and Suggestions for Authors

The authors have presented a robust data set related to virus removal from dates. There are some issues in clarity related to the use of specific dates. Perhaps the introduction could include information on the consumption of dates and the handling and production of dates. Perhaps information on the epidemiological factors related to the one outbreak or if there are others? The number of outbreaks seems unclear between the introduction and the discussion in line 358 where multiple outbreaks is mentioned without references.  Was there epidemiological evidence linked to the water of pitting machines, as mentioned in lines 441-442?

Just curious about the use of the term “dates”. Is there a Latin name of the fruit that should be used to ensure that around the world individuals know what food type this is? Medjool dates are mentioned in line 69. If these are the only dates tested, then this should be clear in title and in abstract. In line 177 an Iran variety of dates is mentioned. This reinforces the need for specificity.  In the discussion starting in line 305 multipole date types are mentioned.

The type date remains unclear in parts of the manuscript, and is addressed in line 240, but this should be clarified prior to this part of the manuscript. Please clarify if the dates were purchased as pitted or if they were pitted in the lab before or after inoculation. Related to this, whole Medjool dates are mentioned in Table 3. So are these whole dates unpitted? How does the pit relate to any loss of virus?

Perhaps a table in the methods could be useful to clarify types of dates used in each of the systems. There are tables with the results, but it remains unclear as to how the authors went about setting up these experiments originally.

Likewise what are the differences between these dates? How are the surfaces different which inhibit the virus removal? Are there sugar or carbohydrate differences that lead to differences in inhibitors perhaps.

The abstract lacks data. Numerical information should be added including the n values and appropriate statistics performed.

In line 33, the illnesses were in individuals who had both NoV and HAV infections together or when total numbers for individuals infected with either.

Lone 36 is missing a close parentheses.

The titles for the three columns in Figure 1 should be on the top rather than on the bottom.

Is there a need to add tween or anything like that given the inherent stickiness of dates?

In Table 1, is the reference to berries and leafy greens associated with methods an error or was this intentional to compare to other methods?

In Table 2, is ISO c only used given the results from Table 1? This is not clear. This is mentioned but not specifically in lines 71-72.

Are the materials used in the ISO-modC widely available? Should this be mentioned or is it possible to swap out specific materials?

Author Response

The authors have presented a robust data set related to virus removal from dates. There are some issues in clarity related to the use of specific dates.

Comments 1:

Perhaps the introduction could include information on the consumption of dates and the handling and production of dates.

Response 1:

We would like to thank this reviewer for the through review of our manuscript and providing insightful comments. This manuscript aimed to test ISO 15216 derived methodologies in extraction of foodborne viruses from dates and is not a comprehensive review article regarding the consumption trends and risk assessment of this matrix.

There is already some information on date processing Page 13 line 431. We added the following information in the introduction.

Page 2, Line 52. Date production and processing can introduce contamination through manual handling or the use of contaminated water19. Processing can include steps such as cleaning, grading, low-pressure steam heat treatment, sprinkling water and placing under mats in the sun, drying, fumigation, pitting, pasteurization, coating and packing19, 20. However, ready-to eat dates are typically eaten raw or with minimal steam heat treatment during processing19.

Comments 2:

Perhaps information on the epidemiological factors related to the one outbreak or if there are others?

Response 2:

Information regarding four outbreaks associated with dates have been provided with references in the following sentences Page 2 Line 49 to Line 52: “in 2018 two separate multi-national outbreaks have been reported in Denmark and the UK 14, 15, in 2019 an outbreak was reported in Sweden 16, and in 2021 another major HAV outbreak was reported in the UK and Australia 17, 18.”

There is no available information on occurring outbreaks due to the virus detected in the samples tested in this study.

Page 2, Line 82. Until now, no outbreaks associated to those positive date samples has been reported to our organisations.

Comments 3:

The number of outbreaks seems unclear between the introduction and the discussion in line 358 where multiple outbreaks is mentioned without references.  

Response 3:

The text Page 11 line 333 and in the second paragraph of the discussion is referring to the same outbreaks as the introduction. The references were added.

Page 11, Line 333. In recent years, contaminated dates have been implicated in multiple foodborne viral outbreaks, but scarce data are available on the ISO 15216 viral extraction method efficiency for this matrix14-18.

Comments 4:

Was there epidemiological evidence linked to the water of pitting machines, as mentioned in lines 441-442?

Response 4:

The used of water in date pitting machine is an observation. The hypothesis of water as a potential origin of contamination was suggested in the FSANZ report. We added the information on the hypothesis from the FSANZ report.

Page 13, Line 424. Dates can be contaminated by the use of water when rinsing the fruits after harvesting9.

Comments 5:

Just curious about the use of the term “dates”. Is there a Latin name of the fruit that should be used to ensure that around the world individuals know what food type this is?

Response 5:

The Latin name of the palm date plant was added:

Page 2, Line 47.  In recent years, multiple outbreaks of HAV involving hundreds of individuals have been associated with the consumption of imported of palm dates (Phoenix dactylifera) from the Middle East and North Africa.

Comments 6:

Medjool dates are mentioned in line 69. If these are the only dates tested, then this should be clear in title and in abstract.

Response 6:

This is not the case.

Page 1, Line 20 we have indicated: “Finally, we tested the adopted method on various varieties of date including pitted ones…”

As indicated in the introduction Page 2, Line 79: “We then selected the best performing method for extraction of multiple foodborne viruses from whole Medjool and successfully applied the method for extraction and characterization of foodborne viruses from different varieties of dates and naturally contaminated products.”

We also indicated Page 3, Line 107 in materiel and method that the sample represent different date varieties, as well as in table 4, 5, 6 and Supplementary table S2, S3 and in the text, other date varieties were tested.

Comments 7:

In line 177 an Iran variety of dates is mentioned. This reinforces the need for specificity.

Response 7:

We modified the term “fruit variety” to “date varieties” in Page 3, Line 107:

Page 3, Line 107. The type (whole=W-, pitted=P-), the date varieties (Uns=unspecified), the country of origin (Unk=unknown), the brand (A to T), the sample lot (1 to 3 when multiple) and a unique subsample number were used for sample identification (ex. P-Sayer/Iran/H1-277).

Comments 8:

In the discussion starting in line 305 multipole date types are mentioned.

The type date remains unclear in parts of the manuscript, and is addressed in line 240, but this should be clarified prior to this part of the manuscript.

Response 8:

As indicated in the section 2.2. Materials and Methods, both whole and pitted dates were tested.

Page 3, Line 107. The type (whole=W-, pitted=P-), the date varieties (Uns=unspecified), the country of origin (Unk=unknown), the brand (A to T), the sample lot (1 to 3 when multiple) and a unique subsample number were used for sample identification (ex. P-Sayer/Iran/H1-277).

More details on the type of dates were added in the abstract and introduction:

Page 1, Line 16. “…but herein we have compared the efficiency of three different protocols based on the ISO 15216 method in extraction of murine norovirus (MNV) from whole Medjool dates and successfully employed the best performing method in extraction of HAV, HuNoV GI and GII and determined the limit of detection (LOD95) of 61, 148, 184 genomic equivalent (gEq) per 25 g, respectively. Finally, we tested the adopted method on various varieties of date including pitted ones (...)”

Page 2, Line 76. “Herein, we compared the efficiency of two different protocols within the ISO 15216 method for extraction of MNV from whole Medjool dates; one protocol that is specified for viral extraction from soft fruits, and the other for viral extraction from leafy greens. We also evaluated the efficiency of two different RNA extraction kits. We then selected the best performing method for extraction of multiple foodborne viruses and successfully applied the method for extraction and characterization of foodborne viruses from different types and varieties of dates and naturally contaminated products.”

Comments 9:

Please clarify if the dates were purchased as pitted or if they were pitted in the lab before or after inoculation. Related to this, whole Medjool dates are mentioned in Table 3. So are these whole dates unpitted? How does the pit relate to any loss of virus?

Response 9:

We added in method that the dates were purchased as whole or pitted.

Page 3, Line 103. Subsamples were prepared with date samples, whole or pitted, collected in various local stores in Quebec.

Comments 10:

Perhaps a table in the methods could be useful to clarify types of dates used in each of the systems. There are tables with the results, but it remains unclear as to how the authors went about setting up these experiments originally.

Response 10:

In the introduction and abstract we indicated that the method was developed with the whole medjools and then applied to different types and varieties of date.

Comments 11:

Likewise what are the differences between these dates? How are the surfaces different which inhibit the virus removal? Are there sugar or carbohydrate differences that lead to differences in inhibitors perhaps.

Response:

Identifying the biochemical differences that exist between these dates are beyond the expertise of the authors and beyond the scope of the journal. We can only speculate that there are differences in the carbohydrate and protein content of these varieties but we cannot confirm those through experiments or literature review. In the discussion we have point out some of the differences that could impact the virus recovery:

Page 12, Line 378. Accordingly, the MNV average recovery yields were lower with the pitted compared to the whole dates but also greatly varied between different varieties. This could be explained by the broken skin of the pitted dates, which leads to exposure of high-carbohydrate inner parts of the dates that could complicate the viral extraction and detection. Furthermore, the type of date, aging and drying process could influence the level of potential inhibitors and thus the recovery rates.

Page 12, Line 386. Furthermore, additives such as sunflower oil was present on the ingredient label of some pitted-dates varieties which could negatively impact the recovery rates. Thus, the phenol chloroform extraction step could improve the recovery rates for those pitted dates, but this specific use and its impact was not tested in this study.

Comments 12:

The abstract lacks data. Numerical information should be added including the n values and appropriate statistics performed.

Response 12:

We added relevant numerical information on the LOD and number of positive naturally contaminated varieties detected in the abstract.

Page 1, Line 15. There is no standard method for extraction and detection of foodborne viruses from dates, but herein we have compared the efficiency of three different protocols based on the ISO 15216 method in extraction of murine norovirus (MNV) from whole Medjool dates and successfully employed the best performing method in extraction of HAV, HuNoV GI and GII and determined the limit of detection (LOD95) of 61, 148, 184 genomic equivalent (gEq) per 25 g, respectively. Finally, we tested the adopted method on various varieties of date including pitted ones and reported the detection of HuNoV GI and GII from 4 naturally contaminated dates varieties.

Comments 13:

 In line 33, the illnesses were in individuals who had both NoV and HAV infections together or when total numbers for individuals infected with either.

Response 13:

It represents the total numbers for individuals infected with either.

Page 1, Line 32. Over 14% of all foodborne outbreak-related hospitalisations reported in the EU were caused by HuNoV or HAV 10.

Comments 14:

Lone 36 is missing a close parentheses.

Response 14:

Page 1, Line 37. The closing parenthesis was added.

Comments 15:

The titles for the three columns in Figure 1 should be on the top rather than on the bottom.

Response 15:

The titles for the three columns in Figure 1 were moved to the top.

Comments 16:

Is there a need to add tween or anything like that given the inherent stickiness of dates?

Response 16:

We wanted to evaluate the ISO 15216 protocol as indicated Page 2, Line 76. We have not tested the tween with the ISO 15216 protocols, but the tween might interfere with the peg precipitation.

Comments 17:

In Table 1, is the reference to berries and leafy greens associated with methods an error or was this intentional to compare to other methods?

Response 17:

The terms used in ISO 15216 are soft fruits and vegetables. The terms in Table 1 and introduction were corrected.

Page 2 Line 76. Herein, we compared the efficiency of two different protocols within the ISO 15216 method for extraction of MNV from whole Medjool dates; one protocol that is specified for viral extraction from soft fruits, and the other for viral extraction from vegetables.

Comments 18:

In Table 2, is ISO c only used given the results from Table 1? This is not clear. This is mentioned but not specifically in lines 71-72.

Response 18:

This is described in the title of table 2 titles. The typography was corrected.

Page 6, Line 220. Table 2. HuNoV and HAV RT-qPCR recovery rates from whole Medjool dates using the ISO-modC extraction protocol.

Comments 19:

Are the materials used in the ISO-modC widely available? Should this be mentioned or is it possible to swap out specific materials?

Response 19:

The ISO 15216 materials are widely available and used in multiple laboratory as well as Qiagen company kits.

Round 2

Reviewer 2 Report

Comments and Suggestions for Authors

The authors have responded well to the comments by the reviewers and improved the manuscript.